# Validation of a new SAFRAN-based gridded precipitation product for Spain and comparisons to Spain02 and ERA-Interim.

Pere Quintana-Seguí[1], Marco Turco[2], Sixto Herrera[3], and Gonzalo Miguez-Macho[4]

[1]Observatori de l'Ebre, (OE), CSIC – Universitat Ramon Llull. Horta Alta 38, 43520 Roquetes, Spain.
[2]Barcelona Supercomputing Center-Centro Nacional de Supercomputación (BSC-CNS), Carrer Jordi Girona 31, 08034 Barcelona, Spain
[3]Meteorology Group, Department of Applied Mathematics and Computer Science, Universidad de Cantabria, Avenida de los Castros s/n, 39005 Santander, Spain
[4]Grupo de Física Non Lineal, Universidade de Santiago Compostela, Calle Xosé María Suárez Núñez s/n., 15782 Santiago de Compostela, Spain

*Correspondence to:* Pere Quintana-Seguí (pquintana@obsebre.es)

**Abstract.** Offline Land-Surface Model (LSM) simulations are useful for studying the continental hydrological cycle. Because of the nonlinearities in the models, the results are very sensitive to the quality of the meteorological forcing; thus, high-quality gridded datasets of screen-level meteorological variables are needed. Precipitation datasets are particularly difficult to produce due to the inherent spatial and temporal heterogeneity of that variable. They do, however, have a large impact on the simulations, and it is thus necessary to carefully evaluate their quality in great detail.

This paper reports the quality of two high-resolution precipitation datasets for Spain at the daily time scale: the new SAFRAN-based dataset and Spain02. SAFRAN is a meteorological analysis system that was designed to force LSMs and has recently been extended to the entirety of Spain for a long period of time (1979/80-2013/14). Spain02 is a daily precipitation dataset for Spain and was created mainly to validate Regional Climate Models. In addition, ERA-Interim is included in the comparison to show the differences between local high-resolution and global low-resolution products. The study compares the different precipitation analyses with rain gauge data and assesses their temporal and spatial similarities to the observations.

The validation of SAFRAN with independent data shows that this is a robust product. SAFRAN and Spain02 have very similar scores, although the later slightly surpasses the former. The scores are robust with altitude and througout the year, save perhaps in summer, when a diminished skill is observed. As expected, SAFRAN and Spain02 perform better than ERA-Interim, which has difficulty capturing the effects of the relief on precipitation due to its low resolution. However, ERA-Interim reproduces spells remarkably well, in contrast to the low skill shown by the high-resolution products. The high-resolution gridded products overestimate the number of precipitation days, which is a problem that affects SAFRAN more than Spain02 and is likely caused by the interpolation method. Both SAFRAN and Spain02 underestimate high precipitation events, but SAFRAN does so more than Spain02. The overestimation of low precipitation events and the underestimation of intense episodes will probably have hydrological consequences once the data are used to force a land surface or hydrological model.

## 1 Introduction

A good knowledge of the continental water cycle and its quantification is critical for society, mainly because it represents both a resource and a hazard. As a consequence, a better understanding of the continental water cycle is a key problem for climate change impact/adaptation studies, and it has become a strategic topic in national and international climate programs such as the Global Energy and Water Cycle Exchanges Project (GEWEX), at the global scale, or the Hydrological Cycle in the Mediterranean Experiment (HyMeX), at the Mediterranean scale. Simulating the continental water cycle is difficult. For example, precipitation is a challenging variable due to its spatial heterogeneity and variability and its time variability. Evapotranspiration is a very complex process that is difficult to measure and depends on the atmosphere, soil and vegetation in a complex manner (e.g., Peters-Lidard et al., 2011; Nasonova et al., 2011; Long et al., 2014).

There are different approaches to simulating the continental cycle. These range from easy-to-calibrate simple water balance models (see, e.g., Orth and Seneviratne, 2015) to the more complex Land-Surface Models (LSM). LSMs simulate the physical processes at the interface among the soil, vegetation and atmosphere. These models may be run offline, forced by a gridded dataset of screen-level meteorological variables, or online and coupled to an atmospheric model.

The main advantage of using offline LSMs is that they avoid atmospheric model biases because they are forced by gridded observational datasets, but they miss many feedback processes. Such systems have been applied at the global, continental, national and basin scales (Rodell et al., 2004; Decharme et al., 2012; Balsamo et al., 2012; Cosgrove, 2003; Mitchell, 2004; Chen et al., 2007; Habets et al., 2008; Artinyan et al., 2008; Szczypta et al., 2012; Barbu et al., 2014; Martínez de la Torre, 2014) in a wide range of works that include the study of water resources, the initialization of meteorological models and the study of the continental water cycle.

The performance of offline LSM simulations depends strongly on the quality of the meteorological datasets used to force the models. In addition, the non-linear nature of the LSMs, which may amplify the errors, makes it essential to ensure the good quality of the meteorological forcing datasets for land surface and hydrological modeling. Unfortunately, it is not an easy task to build such gridded datasets that include all of the meteorological parameters and reach the spatial (at least 5 km) and temporal (intra-daily) resolutions needed to force an LSM. Meteorological analysis systems such as CANARI (Taillefer, 2002), SPAN (Rodríguez et al., 2003; Navascués et al., 2003; Cansado et al., 2004), MESAN (Häggmark et al., 2000) MESCAN (Soci et al., 2016) or SAFRAN (Durand et al., 1993, 1999; Quintana-Seguí et al., 2008; Vidal et al., 2010; Quintana-Seguí et al., 2016) are well suited for this task.

Concerning precipitation, which is the variable examined in this paper, there are other interesting reference datasets, which might not be able to force a LSM because they may not have sufficient temporal (they are often daily products) or spatial (they have resolutions that range from 50 to 12 km) resolution; however, they are well known and widely used for other applications such as the validation (see, e.g., Gómez-Navarro et al., 2012; Turco et al., 2013a) and post-processing of Regional Climate Models (Turco et al., 2011; Casanueva et al., 2015) or the statistical downscaling of Global Climate Models (Casanueva et al., 2016).

It is interesting to compare the analysis systems mentioned in the previous paragraph with these daily reference datasets. At the European scale, E-OBS (Haylock et al., 2008) is probably the best known of such products. However, in some countries the station density considered when building E-OBS was very low, thus limiting the capability of that dataset to properly reproduce the climatic variability of those regions. As a result, some products have been developed for specific regions or countries, such as the gridded dataset *EURO4M-APGD* (Isotta et al., 2014) for the Greater Alpine Region and Spain02 (Herrera et al., 2012, 2016) in Spain, which considers a denser station network.

Careful validation of these products should be performed to ensure that they correctly reproduce the important characteristics of the climate of the region under study. The validation should test both the temporal and spatial accuracies of the products not only for the mean values but also for extremes. Some examples of such studies are Ensor and Robeson (2008), who analyzed the statistics of precipitation in gridded products in the USA and found that the interpolation process significantly increased the number of low-precipitation events while greatly reducing the intensity and frequency of heavy-precipitation episodes; Prein and Gobiet (2016), who evaluated many regional datasets in Europe and demonstrated that the differences among them have the same magnitude as the precipitation errors found in Regional Climate Models; Flaounas et al. (2012), who validated the E-OBS dataset in the Mediterranean region and found that those data must be used with care in complex environments such as coastal areas and mountainous regions; Turco et al. (2013b), who compared three different gridded datasets in the great alpine region (E-OBS and two local grids) and found that specific regional products are often more trustworthy than E-OBS; Herrera et al. (2012), who compared Spain02 and E-OBS in Spain and found that the station density severely affects the results; Turco and Llasat (2011), who confirmed the high quality of the Spain02 dataset in northeast Spain; Belo-Pereira et al. (2011), who compared IB02 - an Iberian daily precipitation dataset built by joining two methodologically equivalent gridded products for Portugal (PT02) and Spain (Spain02-v2) - to global datasets and found that the global products produce better results in Western Iberia than on the Mediterranean side; and Katsanos et al. (2015), who validated the CHIRPS global remote sensing product (Funk et al., 2015) in Cyprus and compared it to E-OBS and local rain gauge data and found that at the monthly scale, the results showed a good correlation between the CHIRPS values and recorded precipitation.

The objective of this paper is twofold. Firstly, this papers validates SAFRAN's precipitation, as reproduced in the most recent Spanish SAFRAN dataset (Quintana-Seguí, 2015), which is a temporal and spatial extension of the dataset presented in Quintana-Seguí et al. (2016). The new dataset spans a period of 35 years (1979/80-2013/14) and geographical covers mainland Spain and the Balearic Islands. Secondly, this paper compares the SAFRAN precipitation dataset with Spain02, which is a well-known, high-quality daily precipitation dataset for Spain, and ERA-Interim, ECMWF's reanalysis dataset.

ERA-Interim is not the same kind of dataset as SAFRAN or Spain02, in the sense that it is a reanalysis product where gauge observations are not assimilated. Precipitation in ERA-Interim is the direct result of a model forecast, even though the dynamics conducent to it are indeed constrained by other assimilated observational data in the reanalysis procedure. Nevertheless, due to its high quality, ERA-Interim is of special interest in hydrology because it is the basis for other derived products that are tailored for hydrological applications such as WFDEI (Weedon et al., 2014). Moreover, ERA-Interim is a dataset suitable to force LSMs, so it is introduced in the comparison to show how a low-resolution global product compares to local higher-resolution ones, which is one of the objectives of the eartH2Observe project and, thus, of this paper. In the case of E-OBS, it

has not been included for several reasons. On the one hand, E-OBS can be considered a different kind of product in comparison to SAFRAN and Spain02 due to its much lower density of stations and the lower resolution of the final product. On the other hand, E-OBS's station density is especially low in Spain; as a consequence, this dataset cannot correctly reproduce Iberian precipitation regimes, which primarily affects the variability and extremes of precipitation events that are well-known problems reflected in previous studies (e.g., Herrera et al., 2012). Finally, contrary to ERA-Interim, it is not suitable to force LSMs, which is the main objective of SAFRAN.

## 2   Study Area

[Figure 1 about here.]

The geographical domain of this study is the peninsular Spain and the Balearic Islands, located in southwestern Europe. Two main factors modulate the climate in this area, the influence of both the Atlantic Ocean and the Mediterranean Sea, and the complex orography (Fig. 1 a). On the one hand, frontal systems from the Atlantic Ocean swipe the region regularly bringing wet and cold air, whilst wet and warm air arrives from the Mediterranean Sea. On the other hand, the complex orography and the configuration of the mountain ranges drive the wet air masses distributing the precipitation across the region. The combination of both factors leads to a marked NW-SE precipitation gradient ranging from large amounts of precipitation (from 900 to 2500 $\mathrm{mmy}^{-1}$) without dry season in the North Atlantic coast to semi-arid and desert regions in the southeastern with less than 100 $\mathrm{mmy}^{-1}$ concentrated in some severe events, with variants of Atlantic and Mediterranean climates in between (AEMET, 2011).

The impact of a surrounding water-mass and a complex orography is particularly relevant in the Ebro valley, in the northeast of Spain. With both Atlantic and Mediterranean influences, the region has a low mean precipitation that is caused by the shadowing effect of the surrounding orography. This is an example of the important role of Spain's marked relief on the distribution of precipitation, enhancing it in some areas and decreasing it in others, which in turn has an important effect on runoff generation. This is an added difficulty for gridded precipitation products. Panel (a) of Fig. 1 shows the area of study, its relief and the main river basins.

## 3   Datasets and Methods

### 3.1   SAFRAN

The SAFRAN (*Système d'Analyse Fournissant des Renseignements Atmosphériques à la Neige*) meteorological analysis system (Durand et al., 1993, 1999) was initially created with the objective of forcing the CROCUS snow model (Brun et al., 1989) in the French Alps to improve avalanche prediction. Currently, it is extensively used in a wide range of applications. SAFRAN uses an Optimal Interpolation algorithm (OI) (Gandin, 1966), which combines observations and a first guess (e.g., the out-

puts of a meteorological model) to produce a gridded dataset of precipitation, temperature, wind speed, relative humidity and cloudiness. SAFRAN is also able to calculate downward visible and infra-red radiation (Ritter and Geleyn, 1992).

SAFRAN is currently used for operational and research purposes in France (Quintana-Seguí et al., 2008; Vidal et al., 2010; Habets et al., 2008). It is used for research purposes in Spain (Quintana-Seguí et al., 2016) and Morocco (Szczypta et al., 2015).
In Spain, SAFRAN was first implemented and tested in a pilot study for the northeastern Iberian Peninsula using only one year of data (Quintana-Seguí et al., 2016). This study is based on the recent extension of SAFRAN to the whole of Spain and to a period of 35 years (1979/80-2013/14) (Quintana-Seguí, 2015).

One of the main characteristics of SAFRAN is the use of climatically homogeneous zones to divide the space into analysis areas (panel (b) of Fig. 1). These zones have irregular shapes and cover an area that is generally smaller than 1000 $km^2$.
Ideally, they should have no horizontal gradient, because, within the zone, SAFRAN cannot reproduce it. As it is impossible to design zones without horizontal gradients, the design of the zones should minimize them as much as possible, especially for precipitation. The zones have several vertical levels spaced at 300 m. The lowest level is on the ground; the other levels are located at 300 m, 600 m, etc. (provided that they are above the ground). The highest level of each zone is the first level above the highest point of the relief. SAFRAN estimates the value of the analyzed variable on each level. These values are subsequently
horizontally interpolated to the regular grid, which is how the data is presented to the users. In our implementation, the grid has a resolution of 5 km (panel (a) of Fig. 1 shows the relief of Spain using the exact same grid as SAFRAN). Each grid point has a location and an altitude, according to the relief, and belongs to a given zone (each zone has, on average, 40 grid points). To interpolate the variable to the altitude of the grid point, the values of the two adjacent vertical levels of the zone are used. A consequence is that, even though zones are initially homogenous, grid points within a zone have different values if they are at
different altitudes. By means of the vertical levels, SAFRAN accounts for variations of precipitation with altitude, using the real gradients at the time of analysis. A downside of the climatically homogeneous zones is that they create artificial discontinuities at the borders of the zones. For precipitation, SAFRAN analyzes daily observations, and all other variables are analyzed every six hours. The data are then time interpolated to the hourly scale using different methods for each variable. For precipitation, relative humidity is currently used to hourly distribute the daily precipitation. For each analysis, SAFRAN uses as much data
as possible, after performing a quality control based on an iterative procedure that compares observed and analyzed quantities at the observation location. The number of stations used in the analysis thus changes with time; this optimizes the quality of the daily precipitation but makes the dataset untrustworthy for trend analysis. We note that in this paper we do not study the hourly distribution of precipitation, but only the quality of daily data. For more details on SAFRAN, please see Quintana-Seguí et al. (2008, 2016).
For this study, SAFRAN has been extended to mainland Spain and the Balearic Islands; for this purpose, a new set of climatological zones has been designed. The new zones, which are shown in panel (b) of Fig. 1, also cover Portugal for a possible future extension. Quintana-Seguí et al. (2016) tested two methods to define the zones, one based on contours of river basins and the other based on the meteorological alert zones used by the Spanish Meteorological State Agency (*Agencia Estatal de Meteorología*, AEMET). The results showed that using meteorological alert zones worked slightly better, but the differences
were small. In other words, SAFRAN is quite robust regardless of the zone map used, provided it has physical sense and the

zones have the right sizes. In order to expand the zone map to the whole of Mainland Spain and the Balearic Islands we first tried to use the same meteorological alert zones, but found that, in general, the zones were too big. As a consequence, we decided to create new zones following a subjective method based on the meteorological alert zones, the basin limits and our expert knowledge. In flat areas, the divisions created were more subjective than on the relief, because there is no clear objective climatic delimitation of the terrain. However, it is precisely in these flat areas where SAFRAN is less sensitive to the shape of the zones, as the horizontal gradients are very weak.

In addition, this newer version of SAFRAN uses ERA-Interim as a first guess for most variables, excluding precipitation (the variable analyzed in this paper), for which the first guess is deduced from the observations. The ERA-Interim dataset is available from 1979, which determines that the start date of the SAFRAN be the same. The meteorological station data come from the AEMET network (see Sec. 3.5). Our dataset finishes in August 2014, which is the date the data was requested to AEMET for this study.

## 3.2 Spain02

Spain02 (Herrera et al., 2012, 2016) is a series of high-resolution daily precipitation and temperature gridded datasets developed for peninsular Spain and the Balearic islands. Version 4 of the product has been produced using the standard EURO-CORDEX (European Coordinated Regional Climate Downscaling Experiment) grids at resolutions of 0.44, 0.22 and 0.11 degrees and different interpolation approaches (Herrera et al., 2016). To compare Spain02 and SAFRAN, we chose the AA-3D version for this work, which considers the orography to be a co-variable (3D) in the interpolation process and it is areal representative (AA). More specifically, the AA-3D version is constructed by calculating the areal average of the interpolation, which is performed by applying thin plate splines (TPS) with the orography at the monthly scale and ordinary kriging (OK) on the daily anomalies. In fact, this is the same method employed by the widely used European scale product E-OBS (Haylock et al., 2008). As a result, both SAFRAN and the version considered for Spain02 are representative of areas and account for relief in the interpolation process. In addition, Spain02 presents the same problems for reproducing trends as described for SAFRAN due to the station network considered (2756 rain gauges), which prioritizes spatial density over temporal consistency.

Thereupon, we will use Spain02, or SP02, to refer to the AA-3D version of Spain02, which has a resolution of 0.11 degrees.

## 3.3 ERA-Interim

ERA-Interim (Dee et al., 2011) is the current global atmospheric reanalysis produced by the European Centre for Medium-Range Weather Forecasts (ECMWF). It covers the period from 1979 onward. ERA-Interim has a spatial resolution of 79 km (T255).

## 3.4 Time periods

[Figure 2 about here.]

Figure 2 shows the time periods covered by the SAFRAN and the Spain02 datasets. The latter covers the period from 1971 to 2010, whereas the current 35-year-long SAFRAN dataset has been produced within two projects and can be divided into three sub-periods, as shown in Fig. 2. The central sub-period (labeled SAFRAN1 in the figure) covers the hydrological years (a hydrological year starts on the 1st of September and finishes on the 31st of August of the next calendar year) from 1995/96 to 2006/07. The other sub-period (SAFRAN2) was developed later in another research project including the remaining years in the series between 1979/80 and 2013/14.

For each of the SAFRAN sub-periods, a set of randomly selected stations was not used to perform the analysis in order to obtain an independent validation dataset. The stations were removed randomnly, but making sure that the spatial separation between the selected stations was larger than a minimum distance, in order to guarantee an even spatial distribution of the stations. Unfortunately, for the SAFRAN1 and SAFRAN2 sub-periods, the same validation datasets were not considered, and a common set of independent stations for the whole 35-year period is thus not available for SAFRAN. Furthermore, Spain02 also has its own set of independent stations. The common set of stations that are independent for both SAFRAN and Spain02 is not useful for validation because there are too few stations that are also poorly spatially distributed. Nevertheless, Spain02 has already been validated with independent data by Herrera et al. (2012, 2016); thus, in this study, the comparisons between SAFRAN and Spain02 are performed using dependent data.

As a consequence, in this study:

1. SAFRAN is validated with independent data for the 1995/96-2006/07 period (SAFRAN1).

2. The other comparisons, which also included Spain02 and ERA-Interim, are performed on a common period (1980-2010, natural years) using dependent stations.

## 3.5    Station data

Panel (b) of Fig. 1 shows the locations of the daily precipitation stations used for this study, which were provided by the Spanish Meteorological State Agency (*Agencia Estatal de Meteorología*, AEMET). The figure only depicts the 1237 stations whose time series have at least 90% of the data, which are the stations that were used for our analysis. The triangles correspond to the independent dataset used to validate SAFRAN for the 1995/96-2006/07 time period (249 stations). The smaller dots correspond to the remaining stations, which are used for all other validations (988 stations).

## 3.6    Comparison measures

The methodology used in this paper is based on the approach used by Turco et al. (2013b), with some modifications to account for the hydrological context in which the SAFRAN dataset has been developed.

[Table 1 about here.]

All of the metrics used in this study compare the products with rain gauge data, considering for each station the nearest gridpoints in terms of the Euclidean distance. The dual nature of precipitation, occurrence and amount, makes it necessary

to explicitly consider both components in the analysis and to include appropriated validation measures (e.g. Roc Skill Area) and indicators (e.g. Wet-days frequency). First, the temporal agreement between the time series and the binary (occurrence/not occurrence of precipitation) sequences were studied using the Relative Mean Absolute Error (MAEr) and the Pearson Correlation (CORR), and the Roc Skill Area (RSA) and its components (HIR: Hit Rate; FAR: False Alarm Rate and CAR: Correct Alarm Rate), respectively. In a second stage, the spatial agreement is studied using the standard precipitation indicators shown in Table 1. All indices are calculated on an annual scale in the common period (see Sec. 3.4), although some indices have also been calculated for each season (PRCPTOT, R1 and RX1D) or at a monthly scale (PRCPTOT and RX1D) for each river basin (see the river basin map in Fig. 1). Finally, the spatial pattern obtained for each index and gridded dataset is compared to the observations using the correlation (CORR), the variability (relative standard deviation, STDEVr), the relative centered root mean square error (CRMSEr) and the relative bias (BIASr). To make the different indices comparable, the "relative" statistics (STDEVr and CRMSEr) have been normalized by dividing both the CRMSEr and STDEVr by the standard deviation of the observations (the reference dataset). Therefore, a STDEVr of 1 means that the standard deviation of the product is the same as that of the observations. The BIASr is relative with respect to the mean of the observations. The Taylor diagram (Taylor, 2001) summarizes these measures on a single plot, which has been produced to show the seasonal statistics.

Note that there are differences between the spatial representativity of the considered datasets, ranging from local (stations) to low-resolution ( 79 $km^2$) areal averages (ERA-Interim). These differences have been taken into account by performing, on the one hand, a comparison among the validation scores of the different gridded datasets with respect to rain gauges and, on the other, by defining global scores based on spatial averages over large enough areas (e.g. river basins). Finally, the aforementioned problem has been considered in the following sections when interpreting and discussing results.

## 4   Results

### 4.1   Validation of SAFRAN using independent data

[Table 2 about here.]

The first step of our analysis is to validate the new SAFRAN dataset for Spain using independent stations for the period 1995/96-2006/07 (SAFRAN1). Table 2 summarizes the results at the annual and the seasonal scale, showing that the CORR, MAEr and RSA are slightly degraded when employing independent stations (not used to perform the analysis), but have nevertheless a rather similar mean and interquartile range, which is an indicator of the robustness of SAFRAN. Seasonally, SAFRAN is quite robust across seasons, being summer the season with the worst performance.

### 4.2   Evaluation of the time similarity

The next step of the analysis is to compare the three studied products (SAFRAN, Spain02 and ERA-Interim) to a common set of observations considering the same period (1980 to 2010).

[Figure 3 about here.]

Figure 3 shows the correlation (CORR), Relative Mean Absolute Error (MAEr) and Roc Skill Area (RSA) between either of the three products and the observations. The SAFRAN and Spain02 correlations are very similar, and the main difference is located on the southeastern coast, where SAFRAN has larger correlations. This similarity is also reflected in the statistics of Table 3, with both products having the same mean annual correlation (0.82) and an almost identical interquartile range (approximately 0.1). As expected, ERA-Interim's correlation is lower. In this case, there is a strong east-west gradient, with the correlations higher in the Mediterranean due to the binary component (rain/no rain) of the precipitation, which is moderately well represented in the reanalysis. Precipitation is a variable with many occurrences of zero, which may suggest the use of the Spearman rank correlation coefficient, instead of Pearson's. In order to verify that both measures give the same qualitative information, we calculated the Spearman rank correlation coefficient and found that it is lower for both SAFRAN (0.73) and Spain02 (0.75). The corresponding interquantile ranges are (0.68/0.78 for SAFRAN and 0.69/0.80 for Spain02). Therefore, all results are shifted when Spearman's coefficient is used instead of Pearson's. Notwithstanding, the width of the interquantile range remains almost the same for SAFRAN and it is narrower by only 0.01 for Spain02; thus we conclude that both Spearman and Pearson provide the same outlook when used to assess the relative performance of either product with respect to the observations.

The results obtained with the correlation are, in general, in agreement with those obtained for the RSA. However, for the latter score Spain02 shows a more homogeneous spatial pattern, with similar values in the southeastern coast and the rest of the area. For ERA-Interim the spatial pattern shown in the correlation is extended for the RSA, with an improvement in the Ebro valley and the southeastern coast.

Concerning the MAEr, again SAFRAN and Spain02 show very similar results, as Table 3 corroborates. Spain02's larger errors are mainly concentrated in the southeast, as seen in the correlation, whereas SAFRAN shows a more random spatial pattern. In the case of Spain02, the affected region (close to Murcia) has the highest station density for the entire peninsula, so the interpolation process obtains an areal averaged precipitation value that differs from the local observations of each particular station. This is more relevant when intense local precipitation events occur, which are very frequent in the Mediterranean region, leading to an overestimation of the interpolated values in the surrounding area. The spatially more distributed nature of the SAFRAN's errors points to a close relation with the position of the zone's borders, which produces a homogeneous spatial distribution over the study area. ERA-Interim's errors are higher and quite homogeneous spatially, with the exception of the Mediterranean coast and an area located in the Basque Country (on the eastern side of the Bay of Biscay).

The boxplots shown in pannels (j) and (l) of figure 3 corroborate the similarity between SAFRAN and Spain02, not only in the mean but also in the spatial variability of the scores, and the worse results and higher spatial variability, in terms of the altitude of the box, of ERA-Interim. Note that there is a remarkable agreement between SAFRAN and Spain02 in all scores related to the binary sequence, reflecting, in addition with the correlation, a very similar temporal structure of both datasets.

Table 3 also shows results by season, clearly indicating that the seasonal behavior of SAFRAN and Spain02 is the same. They both are very robust across seasons with the exception of summer, when the scores are degraded, especially for CORR and MAEr.

Both SAFRAN and Spain02 (the AA-3D version used in this paper) consider the relief in their algorithms, and the same statistics were thus calculated for a subset of high-altitude stations (located at 1000 m or higher). The results, which are shown in Table 3, do not reflect significant differences in terms of both correlation and MAEr between high- and low-altitude stations, with both SAFRAN and Spain02 performing equally well, and only some slight differences are shown for the RSA, more relevant for ERA-Interim. The trend of the errors with altitude for all stations was calculated, and it was not significant in any case, not even for ERA-Interim. However, this result does not account for the fact that high-altitude areas are not well represented by observations because there is a lack of stations at the highest elevations.

### 4.2.1 Evaluation of the spatial similarity by means of precipitation indices

In this section, we evaluate how the different analyzed products are able to reproduce a standard set of precipitation indices. First, the annual cycles of monthly mean and daily maximum precipitation are studied for the eleven river basins shown in Fig. 1. The global statistics of each index are then summarized and analyzed.

[Figure 4 about here.]

Figure 4 shows the annual cycle of the maximum daily precipitation (RX1D) for each of the main Spanish river basins. A first look at the figure shows that, as expected, all products underestimate this index and that ERA-Interim is the dataset with the lowest values of RX1D for most basins. Another interesting result is that Spain02 is closer to the observations than SAFRAN, although they are generally close to each other. With respect to ERA-Interim, despite a general underestimation of RX1D, in some cases, it is closer to the observations (e.g., Duero) than SAFRAN and Spain02 or has higher values than either gridded dataset for some months. This is probably due to the relief surrounding these basins (see Fig. 1), which creates a strong rain shadow effect on its leeward side that cannot be reproduced by ERA-Interim because of the coarse resolution.

Concerning the monthly mean (not shown), SAFRAN and Spain02 are closer, with the latter overestimating the monthly precipitation in the east, southeast and Segura basins, especially in autumn, in agreement with the results described in the previous section.

[Table 4 about here.]

[Figure 5 about here.]

Figure 5 shows how both SAFRAN and Spain02 are able to reproduce the spatial patterns and magnitudes for the three indices, although they overestimate the CWD mostly in the high-relief areas and underestimate RX1D, which is a well-known effect of the interpolation process. Concerning ERA-Interim, it misses the effects of the relief, with a strong north-south gradient for the PRCPTOT and CWD indices and a homogeneous spatial pattern in almost all of the study area for RX1D. ERA-Interim is unable to reproduce the intensity of precipitation, which leads to very low values for both PRCPTOT and RX1D.

In addition to Fig. 5, Table 4 summarizes the scores of the three products for the indices defined in Table 1. Both SAFRAN and Spain02 obtain correlations greater than 0.75, with comparable values, whereas ERA-Interim is unable to reproduce the

spatial pattern of most of the indices, with several correlations below 0.65. In general, the three datasets nevertheless un-
derestimate indices related to the magnitude and intensity of precipitation (PRCPTOT, SDII, RX1D and RX5D), whereas
they overestimate R10, R20 and the frequency of wet days. In addition, there is a common tendency to overestimate CWD
and underestimate CDD. There are some exceptions, however, such as the slight overestimation of SAFRAN and Spain02 of

PRCPTOT or the overestimation of the CDD index given by SAFRAN. Finally, in terms of variability, there is a general under-
estimation for the PRCPTOT, SDII, R20, RX1D and RX5D indices, which is as high as 30% for SDII, and an overestimation
for the rest, mainly for the CWD index.

[Figure 6 about here.]

The scores shown in Table 4 are summarized in the Taylor diagram. Figure 6 shows a summary of the seasonal statistics

of PRCPTOT, R1 and RX1D. In agreement with the results shown in the table, a general pattern in all four diagrams is that
SAFRAN and Spain02 are generally close to each other. In addition, the figures show that in general, the results are quite robust
across seasons. The only exceptions are R1 in summer, when the distance between SAFRAN and Spain02 increases in favor
of the latter, and in autumn, when the distance of RX1D increases, also in favor of Spain02. Another interesting difference is
that in summer, the sign of the bias of PRCPTOT is different for SAFRAN and Spain02. The former has a negative bias, and

15 the latter has a positive one.

## 5  Discussion

A general analysis of the results shows that SAFRAN and Spain02 have very similar scores, although the latter slightly sur-
passes the former. The similarities are very consistent accross the scores used and the seasons analyzed. This means that, in
Spain, SAFRAN's daily precipitation, whose algorithm was designed in the late 20th century, is very close to the precipita-

20 tion estimated by more recent and more specialized state-of-the-art products. As expected, SAFRAN and Spain02 have better
scores than ERA-Interim. The most important limitation of the ERA-Interim reanalysis is its inability to capture the effects of
the relief due to its low resolution, as noted in previous studies (see, e.g., Belo-Pereira et al., 2011), and to reproduce the tem-
poral structure of the precipitation, both in terms of occurrence and amount. However, ERA-Interim reproduces length-spells
remarkably well (although not timely), in contrast to the difficulties of high-resolution products.

The high-resolution gridded products overestimate the number of precipitation days. This is probably caused by the inter-
polation method and affects SAFRAN more than Spain02, likely because of the former's use of climatically homogeneous
zones. SAFRAN generates precipitation for a whole zone once there is a localized precipitation event in some of the stations
within, wrongly assigning the event to unaffected stations. After many occurrences, this process causes an overestimation of
the number of wet days for all stations. Similarly, localized high precipitation events may be missed by SAFRAN because it

tends to average the values of all stations in a zone. SAFRAN may also completely miss events in zones with few stations, in
agreement with Ensor and Robeson (2008). The overestimation of low precipitation events and the underestimation of intense
episodes are likely to have consequences when the data are used to force a land surface or hydrological model, and in the future,
it will be necessary to investigate how the aforementioned problems affect the quality of LSM and hydrological simulations.

The comparison has shown a known problem of SAFRAN, which is the higher errors that are often found at the borders of the zones (Quintana-Seguí et al., 2008, 2016). However, Spain02 also has bugs induced by the algorithm, as shown by the higher errors in the Murcia region (southeast). This shows that each algorithm has its own set of limitations, which need to be well documented and explained to the users of the data.

The comparison between the results obtained using dependent and independent data shows that the procedure used to build SAFRAN is robust. Part of this robustness is due to SAFRAN's algorithm, but it is also due to the high density of stations found in the area. Prein and Gobiet (2016) state that "the information content of a gridded dataset is mainly related to its underlying station density and not to its grid spacing". Because the station density in Spain is so high, the algorithms can leverage a lot of information, producing similar results. Thus, Spain, which is a data-rich country, is a relatively easy target for such algorithms.

Vidal et al. (2010) and Herrera et al. (2016) have shown that station density has an impact on the resulting SAFRAN and Spain02 datasets. In a future study, it would be interesting to see how differently the results of SAFRAN and Spain02 degrade with station density.

    Another interesting result, concerning the robustness of the products, is the similarity of the scores obtained from SAFRAN and Spain02 accross seasons, with the exception of summer. This decrease of performance in summer is expected, due to the

15 small scale of the systems that produce precipitation during this season.

    One limitation of this study is that there was no common set of independent stations for the different periods (see Section 3.4 and Fig. 2), and as a consequence the dependent station datasets used to perform the analysis are not exactly the same. At first glance it might seem that this negatively affects the temporal homogeneity of the product; however the approach in our study minimizes this effect. There is always a trade-off between the number of stations included in an analysis and the homogeneity

of these stations. The approach used in SAFRAN favors station density, not temporal homogeneity, thus, if a station is missing one day, it will not be included in the analysis for that day, but it will be considered when there is data again. As a result, the loss of homogeneity resulting from not excluding the same independent stations for each period is diluted by the fact that, day-to-day, the stations are not exactly the same, together with the relatively small number of independent stations. Temporal homogeneity is not the first priority of SAFRAN, which mitigates the effect of this methodological limitation.

Mountain areas are an exception because the station density in those areas is low and insufficient to reproduce the climatic variability of those regions. Our study has shown that the results of SAFRAN and Spain02 are robust with altitude, as the scores were similar for the stations situated at altitudes higher than 1000 m, but these results do not account for the fact that the mountain areas are not well observed. This has hydrological consequences because high-altitude areas are the generators of most of the basin's runoff and, thus, of most of the water resources used in Spain. Further inspection of these aspects is

necessary in the future. In addition, concerning the relief, it is important to underline that ERA-Interim is unable to reproduce the precipitation regimes of most basins due to its inability to reproduce the effects of the surrounding relief.

## 6 Conclusions

SAFRAN and Spain02 have very similar scores, although the latter generally surpasses the former slightly. As expected, SAFRAN and Spain02 are able to reproduce the main characteristics of precipitation in Spain and perform better than ERA-Interim, which has difficulty capturing the effects of relief on precipitation due to its low resolution. SAFRAN is robust because the scores obtained using dependent and independent data are similar. Furthermore, both SAFRAN and Spain02 are also robust with altitude and accross seasons (except summer, when the scores decrease for both products). Era-Interim reproduces spells remarkably well, in contrast to the low skill shown by the high-resolution products. The high-resolution gridded products overestimate the number of precipitation days, which affects SAFRAN more than Spain02, and is likely caused by the interpolation method. Both SAFRAN and Spain02 underestimate high precipitation events, SAFRAN more so than Spain02.

**Data availability**

Four datasets were used for this paper. The SAFRAN dataset for Spain is available for research purposes from the Mistrals-HyMeX database (Quintana-Seguí, 2015). Spain02 v4 is freely distributed for research purposes at the Santander MetGroup THREDDS CATALOG[1]. ERA-Interim is distributed by the ECMWF on its own website[2]. The observational data were obtained from AEMET using the standard forms available on their website[3].

*Author contributions.* P. Quintana-Seguí applied SAFRAN to Spain. He gathered the necessary data, obtained and modified the SAFRAN code, wrote the necessary scripts to run it properly, defined the climatically homogeneous zones and performed the product quality assessment. M. Turco, P. Quintana-Seguí and Sixto Herrera designed the validation and comparison methodology. M. Turco and Sixto Herrera performed the analysis and generated the figures that show the results. P. Quintana-Seguí wrote most of the text and generated some figures. M. Turco and S. Herrera provided feedback and improvements to the text. S. Herrera helped choose the right version of Spain02 for this comparison and also edited some figures for publication. G. Míguez-Macho provided feedback, which improved the experiment and the document, and helped with the language edditing.

*Acknowledgements.* We are grateful to the French National Centre for Meteorological Research (CNRM UMR3539, Météo-France CNRS) for allowing us to use the code of the SAFRAN analysis system for our studies, to the Spanish State Meteorological Agency (AEMET) for sharing their very valuable observational data with us, and to the European Centre for Medium-Range Weather Forecasts (ECMWF) for making their ERA-Interim product openly available. This is a contribution to the FP7 eartH2Observe project (http://www.earth2observer.eu). This project received funding from the European Union's Seventh Program for research, technological development and demonstration under grant agreement No. 603608. This work has been funded by the Spanish Economy and Competitivity Ministry and the European Regional

---

[1] http://meteo.unican.es/thredds/catalog/Spain_CORDEXgrids/catalog.html
[2] http://www.ecmwf.int/en/research/climate-reanalysis/era-interim
[3] https://sede.aemet.gob.es/

Development Fund through grant CGL2013-47261-R. This work is a contribution to the HyMeX (HYdrological cycle in the Mediterranean eXperiment) program (http://www.hymex.org).

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

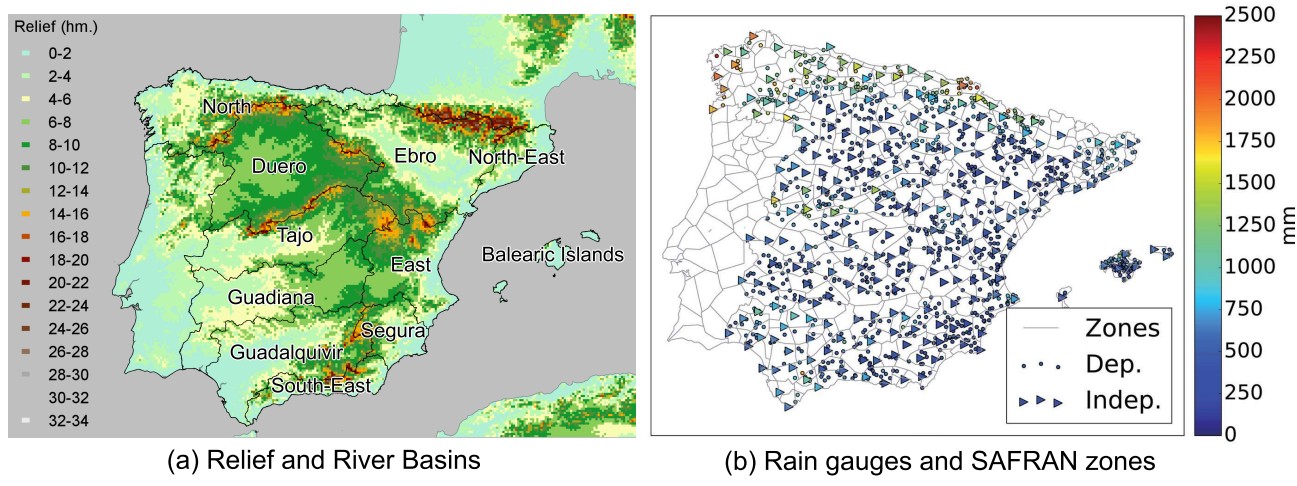

(a) Relief and River Basins            (b) Rain gauges and SAFRAN zones

**Figure 1.** Maps of the study area. The left panel shows the relief (using the same 5 km resolution grid that SAFRAN uses) and the main river basins of Spain (note that for the sake of simplification, some of the smaller river basins have been grouped). The right panel shows SAFRAN's climatically homogeneous zones, the rain gauges used for the study and their mean yearly precipitation. The dependent and independent stations are shown as dots and triangles, respectively.

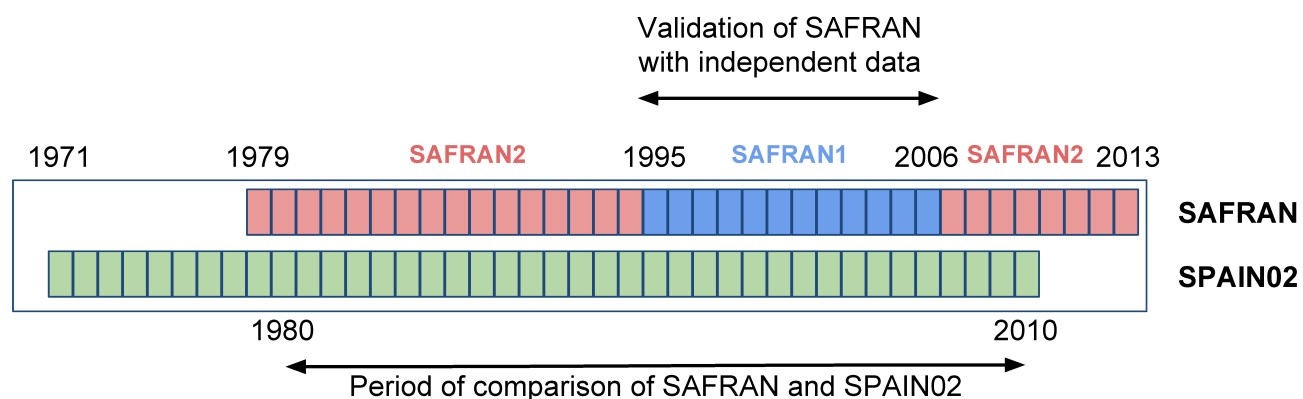

**Figure 2.** Data availability for SAFRAN and Spain02.

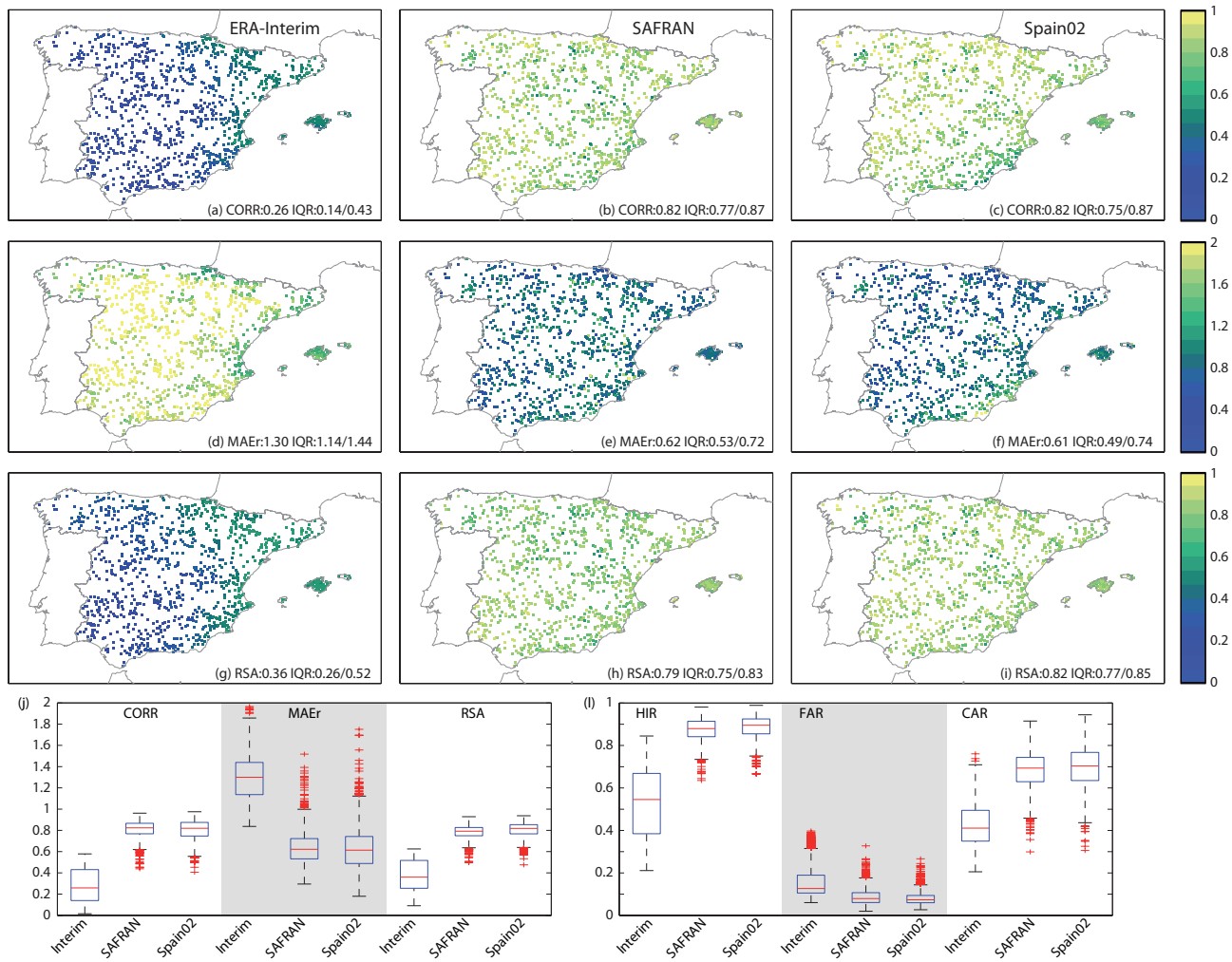

**Figure 3.** Correlation (a-c), Relative Mean Absolute Error (d-f) and Roc Skill Area (g-i) of ERA-Interim, SAFRAN and Spain02 w.r.t. the observations. Lower row shows the boxplots of the (j) CORR, MAEr and RSA, and (l) HIR, FAR and CAR, for the three datasets reflecting the spatial variability of the scores.

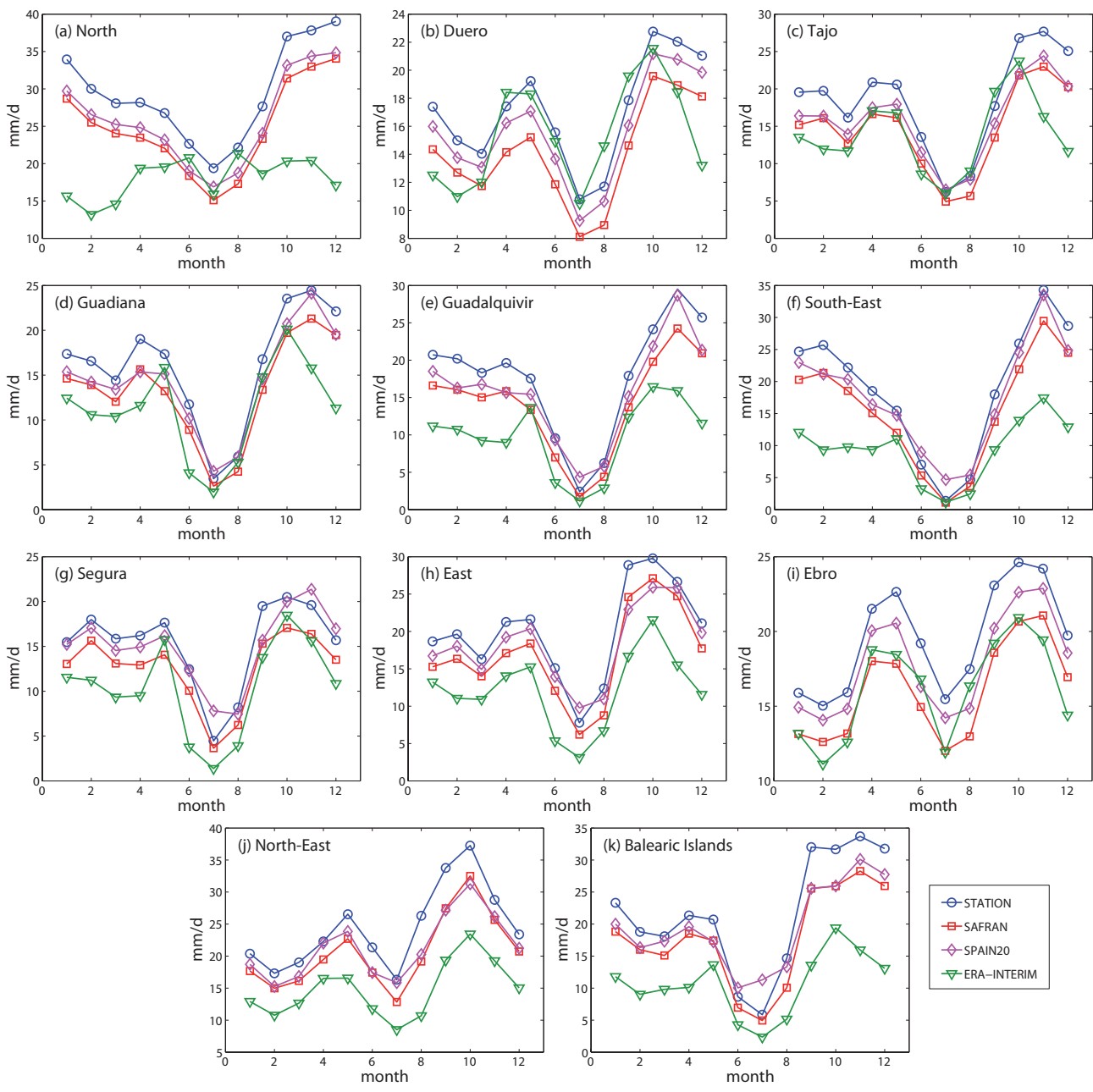

**Figure 4.** Annual cycle of maximum precipitation in one day (RX1D) for the Spanish river basins. The y-axis range is different for each figure to highlight the differences between datasets.

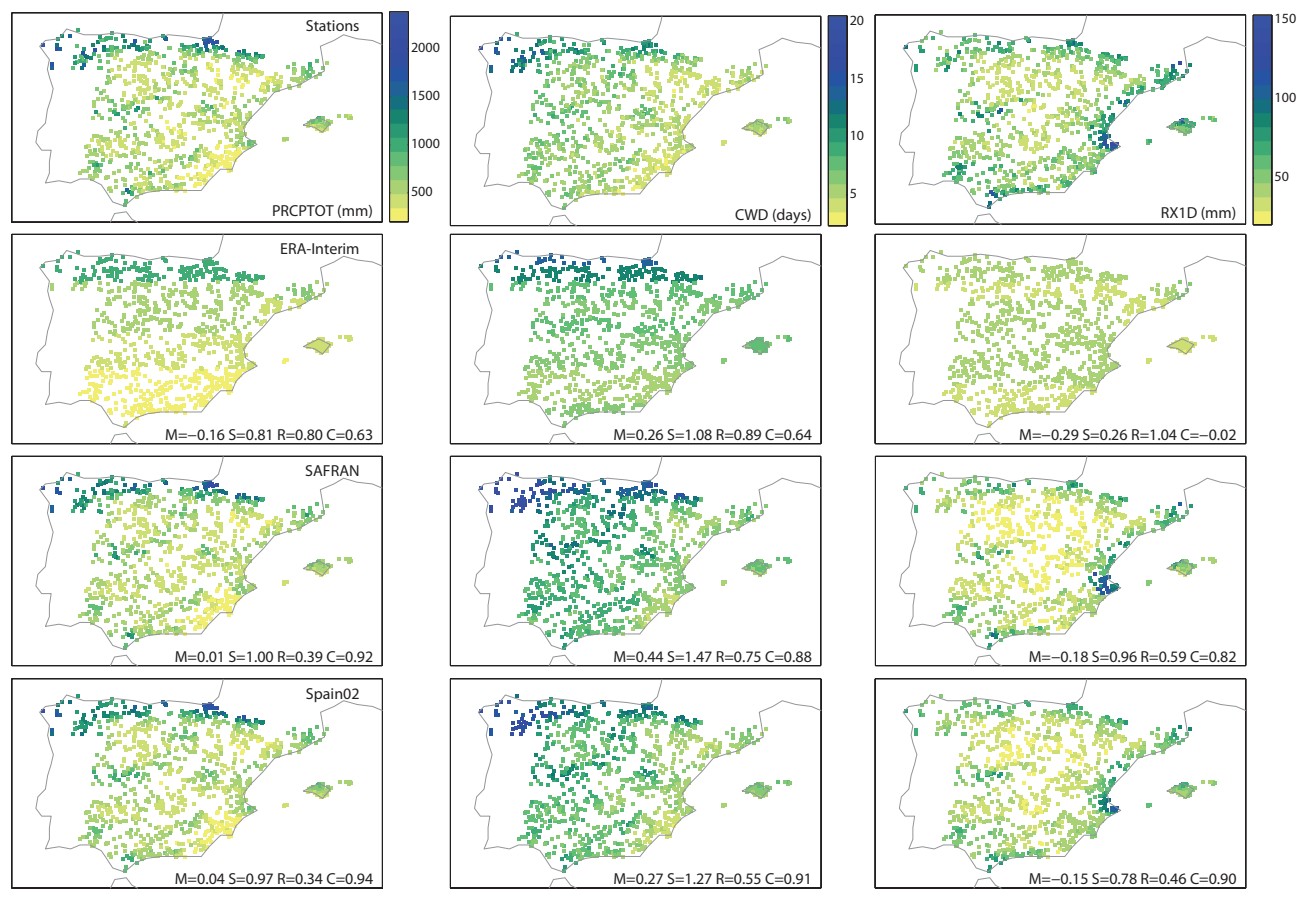

**Figure 5.** PRCPTOT (first column), CWD (second column) and RX1D (third column) for the stations (first row), ERA-Interim (second row), SAFRAN (third row) and Spain02 (fourth row).

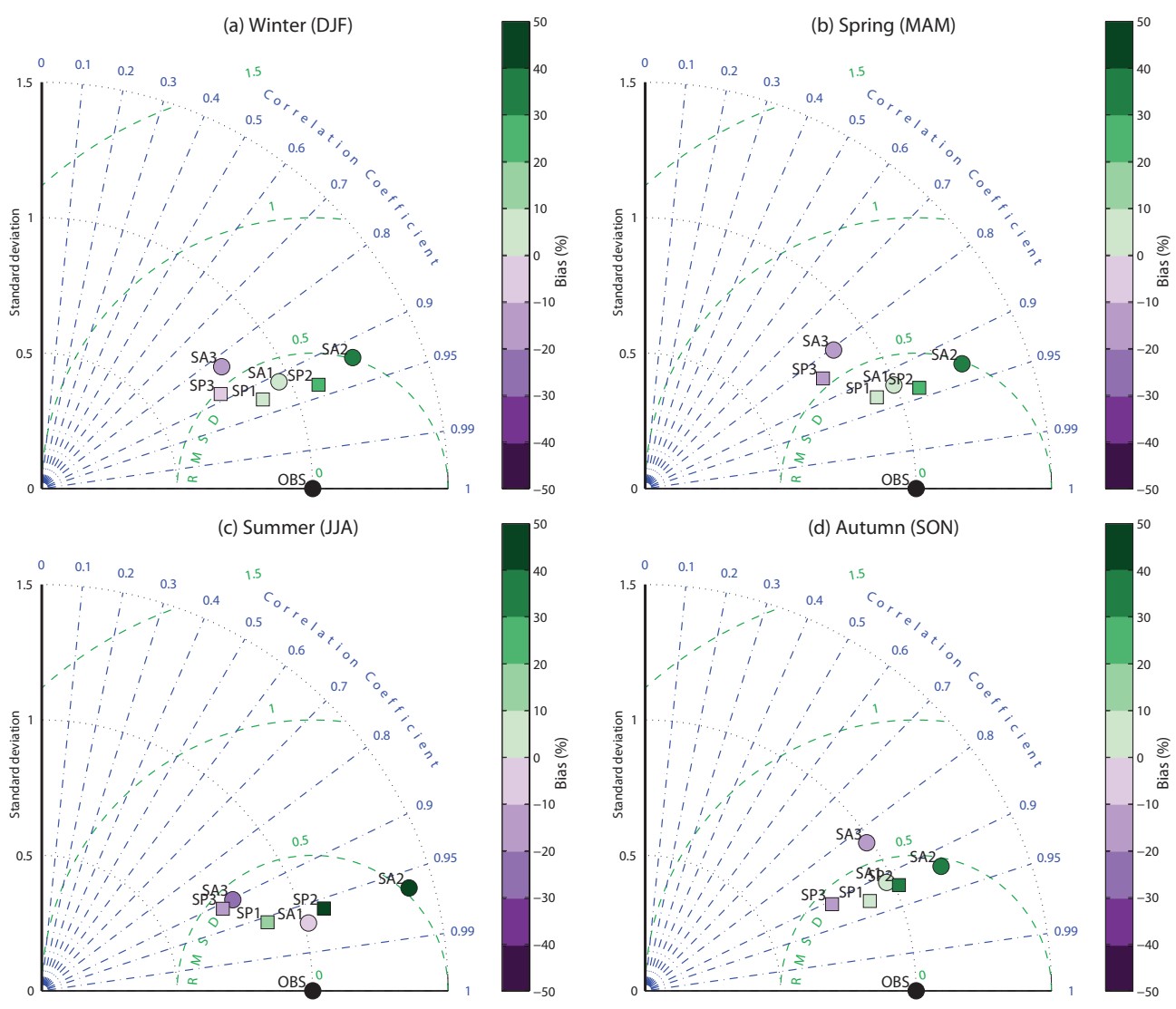

**Figure 6.** Taylor diagrams for seasonal precipitation climatology over Spain. The points most similar to the station data are closest to the point indicated as OBS. Squares labeled "SP" are used for Spain02, whereas circles labeled "SA" are for SAFRAN. The colors indicate the bias (in percentage with respect to the station data) of the dataset. The numbers correspond to the different indices: 1 = PRCPTOT; 2 = R1; and 3 = RX1D.

**Table 1.** Climatic mean and extreme indices for precipitation used in this work, based on ETCCDI (Expert Team on Climate Change Detection and Indices, http://cccma.seos.uvic.ca/ETCCDI).

| Label | Description | Units |
|---|---|---|
| PRCPTOT | total precipitation | mm |
| R1 | number of days with precipitation over 1 $mm.d^{-1}$ (i.e. wet days) | d |
| SDII | mean precipitation amount on a wet day | mm |
| R10 | number of days with precipitation over 10 $mm.d^{-1}$ | d |
| R20 | number of days with precipitation over 20 $mm.d^{-1}$ | d |
| RX1D | maximum precipitation in 1 day | mm |
| RX5D | maximum precipitation in 5 days | mm |
| CWD | consecutive wet days (>1 mm) | d |
| CDD | consecutive dry days (<1 mm) | d |

**Table 2.** Annual and Seasonal Correlation (CORR), Relative Mean Absolute Error (MAEr) and Roc Skill Area (RSA) of SAFRAN compared to independent (Ind.) and dependent (Dep.) observations for the period 1995/96-2005/06.

| | CORR | | | MAEr | | | RSA | | |
|---|---|---|---|---|---|---|---|---|---|
| **Annual** | Mean | Q25 | Q75 | Mean | Q25 | Q75 | Mean | Q25 | Q75 |
| Ind. Obs. | 0.82 | 0.75 | 0.86 | 0.62 | 0.54 | 0.75 | 0.77 | 0.73 | 0.81 |
| Dep. Obs. | 0.82 | 0.77 | 0.87 | 0.62 | 0.53 | 0.72 | 0.79 | 0.75 | 0.83 |
| **Winter** | Mean | Q25 | Q75 | Mean | Q25 | Q75 | Mean | Q25 | Q75 |
| Ind. Obs. | 0.84 | 0.77 | 0.90 | 0.57 | 0.48 | 0.71 | 0.78 | 0.72 | 0.82 |
| Dep. Obs. | 0.85 | 0.79 | 0.89 | 0.58 | 0.48 | 0.69 | 0.79 | 0.74 | 0.83 |
| **Spring** | Mean | Q25 | Q75 | Mean | Q25 | Q75 | Mean | Q25 | Q75 |
| Ind. Obs. | 0.82 | 0.76 | 0.86 | 0.62 | 0.54 | 0.74 | 0.77 | 0.73 | 0.80 |
| Dep. Obs. | 0.83 | 0.78 | 0.87 | 0.61 | 0.53 | 0.71 | 0.78 | 0.74 | 0.82 |
| **Summer** | Mean | Q25 | Q75 | Mean | Q25 | Q75 | Mean | Q25 | Q75 |
| Ind. Obs. | 0.71 | 0.62 | 0.79 | 0.88 | 0.74 | 1.05 | 0.70 | 0.64 | 0.77 |
| Dep. Obs. | 0.74 | 0.66 | 0.80 | 0.82 | 0.71 | 0.94 | 0.75 | 0.70 | 0.80 |
| **Autumn** | Mean | Q25 | Q75 | Mean | Q25 | Q75 | Mean | Q25 | Q75 |
| Ind. Obs. | 0.83 | 0.76 | 0.87 | 0.60 | 0.51 | 0.71 | 0.78 | 0.73 | 0.81 |
| Dep. Obs. | 0.83 | 0.77 | 0.88 | 0.60 | 0.51 | 0.71 | 0.79 | 0.75 | 0.83 |

**Table 3.** Correlation (CORR), Relative Mean Absolute Error (MAEr) and Roc Skill Area (RSA) of the three compared products considering all stations (upper half of the table) and those located at altitudes higher than 1000 m (lower half of the table.

| | Correlation | | | MAEr | | | RSA | | |
|---|---|---|---|---|---|---|---|---|---|
| **Annual** | Mean | Q25 | Q75 | Mean | Q25 | Q75 | Mean | Q25 | Q75 |
| ERA | 0.26 | 0.14 | 0.43 | 1.30 | 1.14 | 1.44 | 0.36 | 0.26 | 0.52 |
| SFR | 0.82 | 0.77 | 0.87 | 0.62 | 0.53 | 0.72 | 0.79 | 0.75 | 0.83 |
| SP02 | 0.82 | 0.75 | 0.87 | 0.62 | 0.49 | 0.75 | 0.82 | 0.77 | 0.85 |
| | | | | >1000 m | | | | | |
| ERA | 0.21 | 0.04 | 0.36 | 1.23 | 1.11 | 1.43 | 0.33 | 0.27 | 0.40 |
| SFR | 0.82 | 0.75 | 0.86 | 0.64 | 0.55 | 0.77 | 0.77 | 0.72 | 0.81 |
| SP02 | 0.82 | 0.75 | 0.88 | 0.61 | 0.49 | 0.76 | 0.80 | 0.73 | 0.85 |
| **Winter** | Mean | Q25 | Q75 | Mean | Q25 | Q75 | Mean | Q25 | Q75 |
| ERA | 0.32 | 0.15 | 0.51 | 1.27 | 1.01 | 1.34 | 0.37 | 0.24 | 0.56 |
| SFR | 0.85 | 0.79 | 0.89 | 0.58 | 0.48 | 0.69 | 0.79 | 0.74 | 0.83 |
| SP02 | 0.83 | 0.76 | 0.89 | 0.58 | 0.45 | 0.70 | 0.81 | 0.76 | 0.85 |
| **Spring** | Mean | Q25 | Q75 | Mean | Q25 | Q75 | Mean | Q25 | Q75 |
| ERA | 0.30 | 0.16 | 0.48 | 1.21 | 1.06 | 1.34 | 0.36 | 0.25 | 0.48 |
| SFR | 0.83 | 0.78 | 0.87 | 0.61 | 0.53 | 0.71 | 0.78 | 0.74 | 0.82 |
| SP02 | 0.82 | 0.76 | 0.88 | 0.59 | 0.47 | 0.71 | 0.82 | 0.77 | 0.86 |
| **Summer** | Mean | Q25 | Q75 | Mean | Q25 | Q75 | Mean | Q25 | Q75 |
| ERA | 0.17 | 0.10 | 0.26 | 1.52 | 1.30 | 1.87 | 0.27 | 0.17 | 0.37 |
| SFR | 0.74 | 0.66 | 0.80 | 0.82 | 0.71 | 0.94 | 0.75 | 0.70 | 0.80 |
| SP02 | 0.72 | 0.57 | 0.81 | 0.89 | 0.67 | 1.30 | 0.80 | 0.73 | 0.85 |
| **Autumn** | Mean | Q25 | Q75 | Mean | Q25 | Q75 | Mean | Q25 | Q75 |
| ERA | 0.23 | 0.10 | 0.40 | 1.37 | 1.13 | 1.55 | 0.36 | 0.25 | 0.53 |
| SFR | 0.83 | 0.77 | 0.88 | 0.60 | 0.51 | 0.71 | 0.79 | 0.75 | 0.83 |
| SP02 | 0.84 | 0.78 | 0.89 | 0.57 | 0.45 | 0.71 | 0.81 | 0.76 | 0.85 |

**Table 4.** Comparison of the performance of ERA-Interim (ERA), SAFRAN (SFR) and Spain02 (SP02) in reproducing indices of mean and extreme precipitation.

| | BIASr | STDEVr | CRMSEr | CORR |
|---|---|---|---|---|
| **Total Precipitation (PRCPTOT)** | | | | |
| ERA | -0.16 | 0.81 | 0.80 | 0.63 |
| SFR | 0.01 | 1.00 | 0.39 | 0.92 |
| SP02 | 0.04 | 0.97 | 0.34 | 0.94 |
| **Consecutive Dry Days (CDD)** | | | | |
| ERA | -0.01 | 1.28 | 0.69 | 0.85 |
| SFR | 0.04 | 1.04 | 0.49 | 0.88 |
| SP02 | -0.03 | 0.91 | 0.50 | 0.87 |
| **Consecutive Wet Days (CWD)** | | | | |
| ERA | 0.26 | 1.08 | 0.89 | 0.64 |
| SFR | 0.44 | 1.47 | 0.75 | 0.88 |
| SP02 | 0.27 | 1.27 | 0.55 | 0.91 |
| **Number of Rainy days (R1)** | | | | |
| ERA | 0.39 | 1.66 | 1.06 | 0.79 |
| SFR | 0.40 | 1.27 | 0.50 | 0.93 |
| SP02 | 0.30 | 1.18 | 0.41 | 0.94 |
| **Mean precipitation of a wet day (SDII)** | | | | |
| ERA | -0.40 | 0.16 | 1.03 | -0.11 |
| SFR | -0.26 | 0.68 | 0.66 | 0.76 |
| SP02 | -0.18 | 0.67 | 0.59 | 0.82 |
| **Number of days of $P > 10$ mm (R10)** | | | | |
| ERA | -0.31 | 0.71 | 0.78 | 0.64 |
| SFR | -0.04 | 1.08 | 0.40 | 0.93 |
| SP02 | 0.05 | 1.06 | 0.34 | 0.95 |
| **Number of days of $P > 20$ mm (R20)** | | | | |
| ERA | -0.57 | 0.25 | 0.94 | 0.38 |
| SFR | -0.22 | 0.94 | 0.46 | 0.89 |
| SP02 | -0.09 | 0.94 | 0.40 | 0.92 |
| **Max. Precip in 1 day (RX1D)** | | | | |
| ERA | -0.29 | 0.26 | 1.04 | -0.02 |
| SFR | -0.18 | 0.96 | 0.59 | 0.82 |
| SP02 | -0.15 | 0.78 | 0.46 | 0.90 |
| **Max Precip in 5 days (RX5D)** | | | | |
| ERA | -0.27 | 0.26 | 1.01 | 0.10 |
| SFR | -0.07 | 0.94 | 0.54 | 0.84 |
| SP02 | -0.06 | 0.85 | 0.42 | 0.91 |