# Peer review of "Validation of a new SAFRAN-based gridded precipitation product for Spain and comparisons to Spain02 and ERA-Interim."

_Hydrology and Earth System Sciences, 2016_

## Referee Comment (RC1) · Anonymous Referee #1 · 13 Oct 2016

Revision of "Validation of a new SAFRAN-based gridded precipitation product for Spain and comparisons to Spain02 and ERA-Interim". This study compares three different daily precipitation datasets in the Peninsular Spain and the Balearic islands. The main purpose of the manuscript is to present a new SAFRAN dataset for the whole Spain and to provide validation metrics. I think that the dataset described in this study is highly relevant given several hydrological applications. The availability of this dataset for scientific purposes is also a valuable output. Nevertheless although the database created is really sound I find that the manuscript could be improved considering two main issues:

i) to include the validation of other variables obtained by the SAFRAN analysis and ii)

to improve the validation of the temporal variability of the data generated.

I include comments on these two issues below. I would recommend the acceptation of the manuscript subject to the major changes suggested below:

1. The first paragraph in the introduction referring to current active research projects is unnecessary. This is not usually explained in scientific publications.

2. Page 3, 25, What is IB02?

3. Page 3, 29-30. The authors indicate that they are presenting a new SAFRAN dataset for the whole Spain. Nevertheless the other objective is less clear. If the authors are using this manuscript to present a whole SAFRAN dataset for the whole Spain including different variables, Why are they only focusing on the validation of precipitation and including a comparison with other precipitation datasets? I would find much more useful to present the new created dataset for the whole Spain as they do for section 3.1, providing more details and then to validate the different variables using observations of the different variables (e.g., wind speed, solar radiation, etc.). This would give much more consistence to the presented research and developed dataset instead of focusing only on the daily precipitation outputs.

4. Section 2. Spatial and temporal variability of precipitation in Spain is very complex, and if authors want to frame the developed dataset on the precipitation characteristics of Spain, they should describe in more depth this complexity. On the contrary, I suggest removing this section; it is not really necessary.

5. Section 3.1: the SAFRAN meteorological analysis system should be described in more depth since this is the basis of subsequent analysis. In particular, the Optimal Interpolation Algorithm should be better explained. Why is the reason of using the period 1979-2014?

6. How are the climatological zones defined here?. I consider this is a key issue that should be described in more depth.

7. Section 3.4. Authors indicate that the SAFRAN dataset was created by two different projects considering different time-spans. Is this approach having some impact on the temporal and spatial homogeneity of the dataset? The approach of considering different dependent and independent station data for validation according to the period of analysis is complex and confuse. Authors should clarify issues of spatio-temporal homogeneity among the two projects.

8. Authors should include more error/accuracy metrics to assess the performance of the gridded data. For example, the correlation coefficient is a very poor measure for temporal agreement between observed and modelled data. I recommend to have a look into the hydroGOF R package.

9. It is not possible for me to read the figure 3. Color scale is not very fortunate. The use of different length for the circles as a function of the values of the validation metrics would be a solution but it would be really useful to show a scatterplot with the metric values between datasets or maybe a boxplot showing the error metrics in the different datasets. A simple average of the correlation coefficients is not very suitable metric to have an idea of the average accuracy of the datasets. Temporal validation would gain if not only the agreement for the entire data is analysed but also temporal agreement for low/high precipitation days, dry spells but also considering possible seasonal influences. For example, it would be useful to know the temporal consistency of the datasets for the different months of the year to determine if the consistency is temporally different between the dry and humid seasons. Also the assessment of the accuracy/error metrics for different elevations would be useful to assess the potential applicability of the data. In this case, the division between stations located above and below 1000 m would be insufficient. Although the number of stations above a certain elevation is low (this is already stated by the authors) it would be very useful to assess the goodness of the prediction in these stations (some of them above 1800 m a.s.l.). Mountain areas are the principal water towers in Spain and where the main floods are generated. For this reason I consider extremely relevant to assess (even using the low

data available) the goodness of the SAFRAN outputs in these regions.

10. Really I would focus in more depth on the validation of the temporal variability of precipitation than on the spatial variability of the average conditions. Usually modelled precipitation tends to reproduce well the average spatial precipitation patterns and the general precipitation seasonality. Thus, given the potential applicability of the SAFRAN dataset to force LSMs, the assessment of the temporal accuracy of the data is much more relevant than the spatial accuracy at the Spanish spatial scale.

11. I find that the discussion section should be improved in more depth including limitations and potentials for the applicability of the SAFRAN dataset considering the proposed analysis related to the temporal precipitation accuracy. Given the potential applicability of the SAFRAN dataset for hydrological applications I find this much more relevant than assessing comparability with the Spain 02 dataset.

---

## Referee Comment (RC2) · Anonymous Referee #2 · 31 Oct 2016

Summary: The paper presents a validation of a precipitation dataset for Spain meant potentially for hydrological and climatological simulations. The new generated dataset spanning more than four decades it is primarily of regional interest. Gridded precipitation data are important for hydro-climatological applications and its evaluation has to be appropriate.

The paper is well structured, the results are clearly presented but the methodology used for verification (Section 3.6) is not sufficiently described. As such, it gives the impression that it is suitable for continuous variables rather than for precipitation. To avoid misleading results for precipitation, the verification should be carried out such as the precipitation datasets used for verification, including rain gauge measurements,

to represent the same spatial scales. I recommend publication once the authors have clarified these aspects and responded to the comments below.

Specific comments:

In order to improve the clarity of the paper, the authors might consider either to expand the acronyms before their first usage (as they have already done it in the first line of the Abstract) or to add an Appendix in which to list all the acronyms in the paper.

Section 1 Page 2, Line 15: Typo: e.g. -> to be removed.

Page 3, Line 33. ERA-Interim precipitation data come from pure low-resolution forecasts and this should be pointed out in the paper.

Section 2. Page 4, Line 14: After Koppen classification a reference is missing and should be introduced.

Section 3.1 Page 5, Line 2: Ritter and Geleyn (1992) developed a scheme for the parametrization of the radiative transfer in numerical weather prediction models. It should be better explained that SAFRAN uses this scheme to produce forecast fields for downward visible and infrared radiation.

Page 5, Line 8: How the climatically homogeneous zones are defined, particularly in areas were no rain gauge measurements are available as in the northeastern Spain (Fig.1b).

Page 5, Line 10: 'The zones have several vertical levels, spaced ... '. How many vertical levels are? Do all zones have the same number of vertical levels?

Page 5, Line 11: 'These values are subsequently horizontally interpolated to a regular grid ...' Does it mean that each zone has its own regular grid? What is the value of the grid-mesh? In addition, how many grid points on a horizontal plane contains a zone? How many analysis points has SAFRAN across Spain? How the analysis horizontal points are defined or chosen?

Page 5, Line 15: 'Afterwards, the data are time interpolated to the hourly scale using different methods for each variable...'. It should be described how accumulated daily precipitation is hourly disaggregated, particularly over the mountains when liquid and solid precipitation may occur during the same day.

Page 5, Line 16: '... SAFRAN uses as much data as possible ...'. How the observation quality control is performed in SAFRAN? For each grid point how many nearby observations are allowed to be used?

Page 5, Line 24: '... for which no first guess is used.' Optimal Interpolation do need a first guess, therefore if no first guess is used for precipitation analysis, what type of interpolation for rain gauge measurements employs SAFRAN? Please describe the precipitation analysis scheme used in SAFRAN.

Section 3.2 Page 6, Line 3: Which is the Spain02 AA-3D grid mesh value used?

Section 3.4 Page 6, Line 13: '... which start in September ... '. Also, it should be mentioned when the hydrological year ends?

Section 3.6 As I have already mentioned in the summary, this section should better describe the methodology used for verification. Unlike gauge measurements which are point observations, model precipitation represents the area of the model grid box, that is about 79 km times 79 km for ERA-Interim but not mentioned for SAFRAN. Comparisons between precipitation observation and the nearest grid point might provide misleading results.

Section 5. Page 10, Line 4 The term 'skill scores' to compare SAFRAN and Spain02 seems unsuitable as throughout the paper no skill scores have been shown. I suggest to use only 'scores' without reference to the skill, both in Conclusions and in the Abstract (page 1, Line 12).

---

## Editor Comment (EC1) · J. Seibert (Editor) · 1 Dec 2016

Thanks to the reviewers for the valuable comments and to the authors for the constructive responses. Based on this discussion, I am confident that the authors will be able to utilize the comments to improve their manuscript.

Reviewer 1 has two major suggestions, including more variables and looking at temporal aspects in more detail. While both are interesting and relevant, I here would tend to agree with the authors response, namely that the former might extend the study too far, but the latter would be a valuable addition, which would clearly strenghten the manuscript.

[Figure]

Re Fig 3, I agre with the reviewer that the color scheme/Scale is subotpimal as differences are hard to see.

I am looking forward to the revised version of this manuscript. Please pay close attention to all the reviewer comments.

Best regards, Jan Seibert

---

## Author Comment (AC1) · 1 Dec 2016

We thank the reviewer for the time spent revising this article and for his/her thoughtful comments, which will help to improve the data analysis and the focus of the article.

Hereafter we will use italics to cite the referee's text and we will use regular type for our own answers.

*Revision of "Validation of a new SAFRAN-based gridded precipitation product for Spain and comparisons to Spain02 and ERA-Interim". This study compares three different daily precipitation datasets in the Peninsular Spain and the Balearic islands. The main purpose of the manuscript is to present a new SAFRAN dataset for the whole Spain*

*and to provide validation metrics. I think that the dataset described in this study is highly relevant given several hydrological applications. The availability of this dataset for scientific purposes is also a valuable output. Nevertheless although the database created is really sound I find that the manuscript could be improved considering two main issues:*

*i) to include the validation of other variables obtained by the SAFRAN analysis and ii)to improve the validation of the temporal variability of the data generated.*

The reviewer raised two main issues. Our opinion on these issues is:

1. Instead of including the validation of other variables, we consider that it is better to focus the article on precipitation only. We will improve the text to highlight more clearly that the focus is entirely on this variable.

2. We agree that the validation of the temporal variability of the data can be improved.

Hereafter we develop the reasoning behind these two opinions and answer, point-by-point the reviewer's comments.

*I include comments on these two issues below. I would recommend the acceptation of the manuscript subject to the major changes suggested below:*

*1. The first paragraph in the introduction referring to current active research projects is unnecessary. This is not usually explained in scientific publications.*

The references to current active projects were introduced to contextualize our work and to underline that these types of datasets have become strategic for national and international climate projects and programs. We prefer to maintain the references for these reasons, but if the reviewer's and editor's point of view is that this does not belong in a scientific paper, we can modify the text and remove those references accordingly.

[Figure]

*2. Page 3, 25, What is IB02?*

IB02 is the union of PT02, a daily precipitation gridded dataset for Portugal, and Spain02-v2 resulting in a merged Iberian daily precipitation dataset. Both components, PT02 and Spain02-v2, were developed using the same methodology ensuring the continuity and spatial homogeneity of the resulting dataset (Belo-Pereira et al., 2011). This will be clarified in the text: "... Spain; Belo-Pereira et al. 2011, who compared IB02 - an Iberian daily precipitation dataset built by joining two methodologically equivalent gridded products for Portugal (PT02) and Spain (Spain02-v2) - to global datasets and found that the global products produce better results in Western Iberia than on the Mediterranean side; ..."

*3. Page 3, 29-30. The authors indicate that they are presenting a new SAFRAN dataset for the whole Spain. Nevertheless the other objective is less clear. If the authors are using this manuscript to present a whole SAFRAN dataset for the whole Spain including different variables, Why are they only focusing on the validation of precipitation and including a comparison with other precipitation datasets? I would find much more useful to present the new created dataset for the whole Spain as they do for section 3.1, providing more details and then to validate the different variables using observations of the different variables (e.g., wind speed, solar radiation, etc.). This would give much more consistence to the presented research and developed dataset instead of focusing only on the daily precipitation outputs.*

We agree with the referee that SAFRAN should be validated for all its parameters/variables. However, the present work builds on the deep analysis and validation already described in Quintana-Seguí et al. (2008), Vidal et al. (2010) and Quintana-Seguí et al. (2016), where such general validation is performed. Thus, a new general validation would not add much new information with respect to previous work. As a consequence, we decided to focus the analysis on a single parameter, precipitation, and to make a more in deep validation of the dataset due to the relevant role of this variable in hydrology studies.

In order to improve the text, we can de-emphasize, in the introduction, the explanations related to other variables, focusing solely on precipitation, in coherence with the rest of the document.

*4. Section 2. Spatial and temporal variability of precipitation in Spain is very complex, and if authors want to frame the developed dataset on the precipitation characteristics of Spain, they should describe in more depth this complexity. On the contrary, I suggest removing this section; it is not really necessary.*

We agree with the reviewer that this section must be improved or removed, as now it does not add a lot of information on the complexity of precipitation in Spain.

*5. Section 3.1: the SAFRAN meteorological analysis system should be described in more depth since this is the basis of subsequent analysis. In particular, the Optimal Interpolation Algorithm should be better explained. Why is the reason of using the period 1979-2014?*

More information about SAFRAN's algorithm could be included in the paper, but our point of view is that this algorithm has already been explained in previous studies such as Durand et al. (1993, 1999), Quintana-Seguí et al (2008) and Quintana-Seguí et al. (2016a). Thus, it would be redundant to reproduce the same explanations in this article.

The period of study, 1979/80-2013/14, depends on data availability. As it is reflected in the manuscript, for all variables, except precipitation, SAFRAN depends on the first guess. The ERA-Interim dataset, our first-guess, starts in 1979, which defines the start date of our period. The end-period of our dataset is defined by the availability of AEMET data. As the data request to AEMET was sent in the second half of 2014, our data series finishes in mid 2014. Thus, the period considered was 1979/80-2013/14. We can rewrite the corresponding section in the text in order to clarify this point.

*6. How are the climatological zones defined here?. I consider this is a key issue that*

*should be described in more depth.*

Climatological zones are an important part of the SAFRAN approach. They should be areas of about 1000 km2 with weak horizontal gradients of the different variables, even though in practice, it is impossible to create zones that perfectly fulfil these requirements.

When we decided to implement SAFRAN in NE Spain we started our tests using river basin limits and AEMET's meteorological alert zones (Quintana-Seguí et al. 2016a). Although the difference was small, better results were obtained when using meteorological warning zones.

When we decided to expand SAFRAN to the whole of mainland Spain and the Balearic islands, we found that, in some regions, the meteorological alert zones of AEMET were too big; thus, we decided to subdivide them. We manually modified these large zones with the aid of a map of river basins and our own expert knowledge. In some areas it was very easy to define limits, just using basin boundaries; however in others, such as flat regions where there are no obvious discontinuities in the values of meteorological variables, the divisions were somewhat arbitrary. Note that, in this particular case, this is not a problem since the horizontal gradients are weak.

To sum up, the method used to define the map of zones shown in Figure 1, combined our own expert knowledge on the local climate, meteorological alert zones, river basin boundaries and the knowledge acquired in our previous study showing that the sensitivity to the limits of the zones is low.

We can include a description of the process in the manuscript.

*7. Section 3.4. Authors indicate that the SAFRAN dataset was created by two different projects considering different time-spans. Is this approach having some impact on the temporal and spatial homogeneity of the dataset? The approach of considering different dependent and independent station data for validation according to the period*

*of analysis is complex and confuse. Authors should clarify issues of spatio-temporal homogeneity among the two projects.*

We acknowledge that this is one of the weak points of this study, but, as we will explain, the impact on the results is very low or non-existent.

Ideally, we would have a high density of stations, with no data gaps and homogeneous time series for the whole period; however, in practice, this is not possible and thus we must find the right trade-off between data quantity, quality and data homogeneity. A very homogenous dataset would use only stations that have minimal gaps in previously homogenized time series. This would certainly ensure homogeneity but result in a low resolution dataset, given the sparsity of high quality data. If we want to include more stations to improve the spatial coverage and thus the spatial quality of the product, then stations with gaps must be used, therefore decreasing temporal homogeneity while improving spatial quality. There is no solution for this problem. One must make a choice.

SAFRAN favours spatial coverage over temporal homogeneity. This is the case of our own application of SAFRAN and also of the original French SAFRAN dataset. In Vidal et al. (2010), who made some comparisons between trends obtained using SAFRAN and homogenized time series, the quantitative differences between SAFRAN trends and the trends of the time series are apparent.

For a given time of analysis (every day for precipitation), SAFRAN ingests all available observations. It does an automatic quality assessment, comparing observations with analyzed values, but the data are not previously homogenized. As a consequence, the number of stations ingested is not the same, day to day. If a station does not have data for a given sub-period, it will not be used until it has data again.

In order to validate SAFRAN with independent data, we set some stations aside, randomly selected and making sure that there is a good spatial coverage. The selection is done automatically by a script at the beginning of the process, and these stations are

never ingested by SAFRAN. As our dataset was created in two different projects, this script was run separately in both projects and thus the stations set aside are not the same.

Regarding the impact of this break in the creation of the dataset on its quality and on this specific study in particular we note that:

1. Concerning the homogeneity of the dataset the impact is very low, because, by construction, the SAFRAN product is already not designed for homogeneity and the number of validation stations is low compared with the number of ingested stations (249 vs 988).

2. Concerning the quality of the validation, we already made the necessary steps in our methodology to avoid any problems. Thus, when we compare the dataset with independent stations we only do so for the SAFRAN-1 subperiod (1995/96-2006/07), which ensure the complete independence of the validation dataset. See Page 6, Line 25. Furthermore, most of the comparisons have been performed with dependent data, because there is no common independent dataset for both SAFRAN and Spain02, as these products were created in different periods of time and by different groups without any coordination (Page 6, lines 20-23).

Furthermore, Table 2 and section 4.1 show that SAFRAN is very robust, that is, the scores obtained when validating with independent stations are very close to those obtained when comparing to dependent data. This is also shown in previous studies.

To sum up, we think that, as SAFRAN is not optimized for temporal homogeneity, the impact of the break in the generation of the dataset is not really relevant. Furthermore, the number of validation stations is low compared with the number of stations ingested by the analysis, which implies that the homogeneity impacts should be low. The results of the validation are unaffected because our methodology did take this problem into account.

*8. Authors should include more error/accuracy metrics to assess the performance of the gridded data. For example, the correlation coefficient is a very poor measure for temporal agreement between observed and modelled data. I recommend to have a look into the hydroGOF R package.*

We can include some modifications to the manuscript in order to explore other metrics, being careful not to introduce redundant information. Thus, for example, we could study the possibility of using the Spearman correlation (rank based), in order to better tackle the effect of having many days of zero precipitation, which may dominate the Pearson correlation coefficient.

*9. It is not possible for me to read the figure 3. Color scale is not very fortunate. The use of different length for the circles as a function of the values of the validation metrics would be a solution but it would be really useful to show a scatterplot with the metric values between datasets or maybe a boxplot showing the error metrics in the different datasets.*

We can consider alternative colour scales or plot types, if the editor deems it necessary, but removing the black border line of the dots would probably be enough in order to make the figure clearer. Furthermore, scatter plots of the scores comparing ERA-Int and SAFRAN with Spain02 can be done too. Finally, it is also possible to add a boxplot with the spatial distribution of the errors.

*A simple average of the correlation coefficients is not very suitable metric to have an idea of the average accuracy of the datasets. Temporal validation would gain if not only the agreement for the entire data is analysed but also temporal agreement for low/high precipitation days, dry spells but also considering possible seasonal influences. For example, it would be useful to know the temporal consistency of the datasets for the different months of the year to determine if the consistency is temporally different between the dry and humid seasons.*

We didn't include this kind of detailed analysis because we wanted to keep the article relatively short, but we could add some more results building on the reviewer's suggestions.

*Also the assessment of the accuracy/error metrics for different elevations would be useful to assess the potential applicability of the data. In this case, the division between stations located above and below 1000 m would be insufficient. Although the number of stations above a certain elevation is low (this is already stated by the authors) it would be very useful to assess the goodness of the prediction in these stations (some of them above 1800 m a.s.l.). Mountain areas are the principal water towers in Spain and where the main floods are generated. For this reason I consider extremely relevant to assess (even using the lowdata available) the goodness of the SAFRAN outputs in these regions.*

We can add another altitude threshold on our analysis of the results pertaining to altitude.

*10. Really I would focus in more depth on the validation of the temporal variability of precipitation than on the spatial variability of the average conditions. Usually modelled precipitation tends to reproduce well the average spatial precipitation patterns and the general precipitation seasonality. Thus, given the potential applicability of the SAFRAN dataset to force LSMs, the assessment of the temporal accuracy of the data is much more relevant than the spatial accuracy at the Spanish spatial scale.*

As we said in a previous answer, we will explore the possibility of including one or more new scores, in order to improve this aspect of our study.

*11. I find that the discussion section should be improved in more depth including limitations and potentials for the applicability of the SAFRAN dataset considering the proposed analysis related to the temporal precipitation accuracy. Given the potential applicability of the SAFRAN dataset for hydrological applications I find this much more relevant than assessing comparability with the Spain 02 dataset.*

We can improve the discussion based on the comments of the referees and the analysis proposed in this document.

---

## Author Comment (AC2) · 1 Dec 2016

We thank the reviewer for the time spent reviewing this article and by raising two main issues, which, when addressed, will improve the quality of the manuscript.

Hereafter we will use italics to cite the referee's text and we will use regular type for our own answers.

*Summary: The paper presents a validation of a precipitation dataset for Spain meant potentially for hydrological and climatological simulations. The new generated dataset spanning more than four decades it is primarily of regional interest. Gridded precipitation data are important for hydro-climatological applications and its evaluation has to*

[Figure]

*be appropriate.*

*The paper is well structured, the results are clearly presented but the methodology used for verification (Section 3.6) is not sufficiently described. As such, it gives the impression that it is suitable for continuous variables rather than for precipitation. To avoid misleading results for precipitation, the verification should be carried out such as the precipitation datasets used for verification, including rain gauge measurements, to represent the same spatial scales. I recommend publication once the authors have clarified these aspects and responded to the comments below.*

The two main issues raised by the referee are:

1. The methodology is not sufficiently described.

2. The verification should be carried out such as the precipitation datasets used for verification, including rain gauge measurements, represent the same spatial scales.

Concerning these issues:

1. We agree that we can improve the quality of the methodology section (section 3.6).

2. We think that we must improve the discussion on the sampling errors in both the methodology section and in the discussion, but we do not think we have to change the methodology.

Hereafter we respond in detail to all the questions raised by the reviewer.

*Specific comments:*

*In order to improve the clarity of the paper, the authors might consider either to expand the acronyms before their first usage (as they have already done it in the first line of the Abstract) or to add an Appendix in which to list all the acronyms in the paper.*

We agree and we can apply any of the solutions proposed.

*Section 1 Page 2, Line 15: Typo: e.g. -> to be removed.*

We do not consider this a typo, as we added it consciously. But we can remove it if the editor deems it necessary.

*Page 3, Line 33. ERA-Interim precipitation data come from pure low-resolution forecasts and this should be pointed out in the paper.*

We can clarify this in the text.

*Section 2. Page 4, Line 14: After Koppen classification a reference is missing and should be introduced.*

Yes. The best reference would be Peel et al. (2007), which is the classification used in AEMET (2011), which we already cite.

Peel, M. C., Finlayson, B. L., & Mcmahon, T. A. (2007). Updated world map of the Köppen-Geiger climate classification. *Hydrology and Earth System Sciences*, *11*, 1633–1644.

*Section 3.1 Page 5, Line 2: Ritter and Geleyn (1992) developed a scheme for the parametrization of the radiative transfer in numerical weather prediction models. It should be better explained that SAFRAN uses this scheme to produce forecast fields for downward visible and infrared radiation.*

This can be done. However, following the recommendations of Reviewer 1, we might remove emphasis on the other variables of SAFRAN, different from precipitation.

*Page 5, Line 8: How the climatically homogeneous zones are defined, particularly in*

*areas where no rain gauge measurements are available as in the northeastern Spain (Fig.1b).*

Climatological zones are an important part of the SAFRAN approach. They should be areas of about 1000 km2 with weak horizontal gradients of the different variables, even though in practice, it is impossible to create zones that perfectly fulfil these requirements. The availability, or not, of stations is not considered when designing the zones. In the case there are no stations in the zone, SAFRAN will use information from stations in neighbouring zones. For variables other than precipitation, the first guess is, of course, also used.

When we decided to implement SAFRAN in NE Spain we started our tests using river basin limits and AEMET's meteorological alert zones (Quintana-Seguí et al. 2016a). Although the difference was small, better results were obtained when using meteorological warning zones.

When we decided to expand SAFRAN to the whole of mainland Spain and the Balearic islands, we found that, in some regions, the meteorological alert zones of AEMET were too big; thus, we decided to subdivide them. We manually modified these large zones with the aid of a map of river basins and our own expert knowledge. In some areas it was very easy to define limits, just using basin boundaries; however in others, such as flat regions where there are no obvious discontinuities in the values of meteorological variables, the divisions were somewhat arbitrary. Note that, in this particular case, this is not a problem since the horizontal gradients are weak.

To sum up, the method used to define the map of zones shown in Figure 1, combined our own expert knowledge on the local climate, meteorological alert zones, river basin boundaries and the knowledge acquired in our previous study showing that the sensitivity to the limits of the zones is low.

We can include a description of the process in the manuscript.

*Page 5, Line 10: 'The zones have several vertical levels, spaced ... '. How many vertical levels are? Do all zones have the same number of vertical levels?*

There is the ground level (at whatever altitude it is) and then levels at 300, 600, . . . m. Of course, only the levels higher than the relief are considered. Similarly, the upper level depends on the relief of the zone. As many levels as necessary are included in order to surpass the altitude of the highest point of the zone.

*Page 5, Line 11: 'These values are subsequently horizontally interpolated to a regular grid ...' Does it mean that each zone has its own regular grid? What is the value of the grid-mesh? In addition, how many grid points on a horizontal plane contains a zone? How many analysis points has SAFRAN across Spain? How the analysis horizontal points are defined or chosen?*

We can clarify the description of the interpolation to the grid in the article, even though this is explained in Quintana-Seguí et al (2008, 2016a). We can add the grid points on Figure 1 and, maybe, a map of the gridded precipitation. This would further clarify this point.

There is a single regular grid. It is a 5 km resolution grid in a Lambert Conformal Conic projection that covers all the Iberian Peninsula and the Balearic Islands. All the land grid points belong to a zone. So, once the analysis is performed for each zone, we have a value of precipitation for each altitude level (one value at the ground, another at 300 m., another at 600 m., and so on). Then we find the value of precipitation for each grid point. Each grid point has an altitude. The precipitation of that grid point will be the linear interpolation of the precipitation of the two closest levels. This way, we go from a value of precipitation for each zone and level, to a value of precipitation for each grid point.

*Page 5, Line 15: 'Afterwards, the data are time interpolated to the hourly scale using different methods for each variable...'. It should be described how accumulated daily precipitation is hourly disaggregated, particularly over the mountains when liquid and*

[Figure]

*solid precipitation may occur during the same day.*

We did not explain this in this article because the article focuses on daily precipitation and, thus, hourly interpolation, which is explained in previous papers, does not affect at all the results.

*Page 5, Line 16: '... SAFRAN uses as much data as possible ...'. How the observation quality control is performed in SAFRAN? For each grid point how many nearby observations are allowed to be used?*

SAFRAN does a quality control of the observations. This is an iterative procedure based on the comparison between observed and analyzed quantities at the observation location (Quintana-Seguí et al. 2008).

*Page 5, Line 24: '... for which no first guess is used.' Optimal Interpolation do need a first guess, therefore if no first guess is used for precipitation analysis, what type of interpolation for rain gauge measurements employs SAFRAN? Please describe the precipitation analysis scheme used in SAFRAN.*

Our writing is indeed confusing. The first guess used by SAFRAN is a field of zero precipitation. This gives better results than using, for example, the precipitation fields of ERA-Interim.

*Section 3.2 Page 6, Line 3: Which is the Spain02 AA-3D grid mesh value used?*

The resolution of Spain02 AA-3D considered was 0.11°.

*Section 3.4 Page 6, Line 13: '... which start in September ... '. Also, it should be mentioned when the hydrological year ends?*

We can mention this. In hour context, hydrological years start on the 1st of September and end on the 31st of August.

*Section 3.6 As I have already mentioned in the summary, this section should better describe the methodology used for verification. Unlike gauge measurements which*

*are point observations, model precipitation represents the area of the model grid box, that is about 79 km times 79 km for ERA-Interim but not mentioned for SAFRAN. Comparisons between precipitation observation and the nearest grid point might provide misleading results.*

We agree with the referee that the local or areal representativeness of the different datasets is an important problem for most verification studies. However, in our opinion all analyses in this study are independent of this problem for two main reasons:

- On the one hand, the article is focused on the intercomparison of the products (SAFRAN and Spain02) using a common dataset, with observations as reference. In this sense, all the conclusions are obtained based on the relative differences between the validation scores obtained for both datasets.

- On the other hand, most of the validation has been performed using spatial averages, which makes negligible the effect of the spatial representativeness.

The case of ERA-Interim is slightly different because of its coarser resolution. Nevertheless, we decided to include it in the study as a reference because it is often used to force high resolution (~5 km) LSMs or hydrological models when high resolution data is not available (this happens in data poor countries, where there are no alternatives). In this case it is interesting to show the difference in performance, including both modelling errors and sampling errors.

Finally, SAFRAN and Spain02 are interpolated products that ingest observations and, thus, it is necessary to validate them using the same kind of data they ingest, as all previous studies validating these products do.

We propose to further discuss this important and interesting issue in the "Datasets and Methods" section of the manuscript, which is not well developed in the current version of the manuscript. We should also introduce some references to this problem in the

discussion section. We do not think it is necessary to, for example, upscale SAFRAN or Spain02 to ERA-Interim's resolution, as this is out of the scope of this study.

*Section 5. Page 10, Line 4 The term 'skill scores' to compare SAFRAN and Spain02 seems unsuitable as throughout the paper no skill scores have been shown. I suggest to use only 'scores' without reference to the skill, both in Conclusions and in the Abstract (page 1, Line 12).*

We can change "skill score" to "score".

––––––––––––––––––––––––––––––

---

## Author Response (AR1)

Dear editor,

Following the comments of the reviewers, a new version of the manuscript has been prepared. A document showing all the changes performed on the text has been attached to the end of this letter.

The main changes to the document have been the increased focus on precipitation and the addition of more analysis in order to better study the temporal aspects of the datasets. To this end, new scores have been introduced and the tables have been expanded.

Hereby we list all the changes that have been made to the document. The line numbers we mention, refer to the version of the manuscript which shows the changes and which is attached to the this letter. We think this will help you locate the alterations.

**Recommendations by Ref. 1.**
- Page 2, Lines 5-14: In the introduction, we have reduced the mentions to the projects related to this work.
- Page 3, Lines 27-28: The description of IB02 has been improved.
- Page 3, Lines 32-35; Page 4, Lines 1-4: In order to better focus the paper on precipitation, emphasys on other variables has been diminished.
- Section 2 has been improved.
- Page 4: Lines 4-8: Improved explanation on ERA-Interim. It is a model which does not use rain gauge data, but it assimilates other observations.
- Page 6, Lines 12-23: We have improved the explanation on how the homogeneous zones were created.
- Page 7, Lines 16-19: The time period of the SAFRAN dataset has been better explained.
- Improvement of Figure 03.
  - The color scale and the dot size used in the maps remains the same, but we have removed the black contour, thereby making them easier to read.
  - We have added box plots, which add new and interesting information.
  - The reviewer suggested scatter plots. We finally did not introduce them, as we considered that they were not useful to better interpret the results. The box plots work better.
  - Page 10, Lines 12-15: The box plots have been commented in the text.
- Improvement of the evaluation of the temporal agreement between product and observations.
  - Page 9, Lines 23-29: The Spearman rank correlation coefficient has been calculated, in order to compare it to the Pearson coefficient and to ensure that the conclusions attained by both are the same. This has been discussed in the text.
  - The Roc Skill Area (RSA) has been introduced, in the tables, together with its components (FAR, HIR and CAR), which are included in the new box plot, in

order to better take into account the dual nature of precipitation (occurrence and amount).
- ○ Page 8, Lines 12-18: The RSA has been commented.
- ○ Page 10, Lines 1-4: Comments on the RSA.
- Seasonal influences
  - ○ Tables 2 and 3 have been expanded, to include seasonal scores.
  - ○ Page 9, Lines 8-11: The seasonal scores have been commented in the text.
  - ○ Page 10: Lines 18-20: Comments on the seasonality of the results.
  - ○ Page 13: Lines 1-3: Discussion on the seasonality of the results.
- Page 13, Lines 4-13: A discussion on the impact of having different datasets for different subperiods has been introduced.
- Unexplained acronyms have been explained.

**Recommendations by Ref. 1, which we did not follow:**
- Ref. 1 suggested to use the hydroGOF package in order to improve our paper. We analyzed the package and we decided not to use it, as we think that the inclusion of the Spearman rank correlation and the Roc Skill Area, suffice.
- Ref. 1 also suggested to include a higher altitude threshold for our analysis on the impact of altitude on the performance of the products. We agreed that this should be done. However, we found that in our dataset there is only one station above 1800 m. and 4 stations above 1500, which have enough data for the analysis. Being the number of stations so low, the results would not be statistically robust, so we decided not to perform this analysis.

**Recommendations by Ref. 2.**
- Page 5, Lines 24-25: We clarified how the SAFRAN altitude levels work.
- Page 5, Lines 27-31: The explanation on the spatial interpolation and the regular grid used has been improved.
- Panel (a) of Figure 1 has been completely redone in order to show the resolution of the grid used.
- Page 6: Lines 3-4: The hourly interpolation of precipitation is explained.
- Page 6, Lines 4-5: How the quality control works in SAFRAN has been clarified.
- Page 6, Lines 13-23: We have improved the explanation on how the homogeneous zones were created.
- Page 6, Line 24-26: How the first guess works for SAFRAN precipitation has been clarified.
- Page 7, Line 6: A mention on the resolution of Spain02 has been included.
- Page 7, Lines 16-17: The definition of hydrological year has been introduced.
- Page 8, Lines 27-31: A discussion on the sampling errors has been introduced.
- The concept "skill score" has been removed.

**Recommendations by Ref. 2, which we did not apply:**
- The description of radiation has not been improved because, following the recommendation of Ref. 1 we focused the paper on precipitation.
- A citation on the Köppen classification has not been introduced, because section 2 was rewritten removing reference to the Köppen classification.

**Other changes**
- The abstract, the discussion and the conclusions have been improved, in order to reflect the changes in the document.

Yours sincerely,

Pere Quintana-Seguí
on behalf of all the co-authors.

[revised manuscript text omitted]

---

## Author Response (AR2)

Roquetes, 5th of March 2017.

Dear editor,

A new version of the manuscript has been prepared, which includes all your suggestions.

Yours sincerely,

Pere Quintana-Seguí
on behalf of all the co-authors.

[revised manuscript text omitted]